# Tensor Normalization and Full Distribution Training

**Wolfgang Fuhl,**[1] **Enkelejda Kasneci,** [1]

[1] Human Computer Interaction, University Tübingen
wolfgang.fuhl@uni-tuebingen.de, enkelejda.kasneci@uni-tuebingen.de

## Abstract

In this work, we introduce pixel wise tensor normalization, which is inserted after rectifier linear units and, together with batch normalization, provides a significant improvement in the accuracy of modern deep neural networks. In addition, this work deals with the robustness of networks. We show that the factorized superposition of images from the training set and the reformulation of the multi class problem into a multi-label problem yields significantly more robust networks. The reformulation and the adjustment of the multi class log loss also improves the results compared to the overlay with only one class as label. LinkToCodeBlind

## Introduction

Deep neural networks are the state of the art in many areas of image processing. The application fields are image classification, semantic segmentation, landmark regression, object detection, and many more. In the real world, this concerns autonomous driving, human-machine interaction, eye tracking, robot control, facial recognition, medical diagnostic systems, and many other areas. In all these areas, the accuracy, reliability, and provability of the networks is very important and thus a focus of current research in machine learning. The improvement of accuracy is achieved, on the one hand, by new layers that improve internal processes through normalizations (Ioffe and Szegedy 2015; Salimans and Kingma 2016; Huang et al. 2017; Qiao et al. 2019; Wu and He 2018; Ulyanov, Vedaldi, and Lempitsky 2016; Huang and Belongie 2017) or focusing on specific areas either on the input image or in the internal tensors (Wang et al. 2017; Hu, Shen, and Sun 2018; Hochreiter and Schmidhuber 1997). Another optimization focus are the architectures of the models, through this considerable success has been achieved in recent years via ResidualNets (He et al. 2016), MobileNets (Sandler et al. 2018), WideResnets (Zagoruyko and Komodakis 2016), PyramidNets (Han, Kim, and Kim 2017), VisonTransformers (Dosovitskiy et al. 2020), and many more. In the area of robustness and reliability of neural networks, there has been considerable progress in the area of attack possibilities on the models (Goodfellow, Shlens, and Szegedy 2014; Madry et al. 2017; Carlini and Wagner 2017;

Kurakin et al. 2016) as well as in their defense (Papernot et al. 2016; Strauss et al. 2017; Pang et al. 2019; He et al. 2017; Tramèr et al. 2017; Sen, Ravindran, and Raghunathan 2020).

## Contribution of this work:

- A novel pixel wise tensor normalization layer which does not require any parameter and boosts the performance of deep neuronal networks.
- The factorized superposition of training images, which boosts the robustness of deep neural networks.
- Using a multi label loss softmax formulation to boost the accuracy of the robust models trained with the factorized superposition of training images.

## Related Work

Since this work deals with two subsets of modern DNNs, we have divided the related work into three subareas. The first subarea is internal normalization in DNNs, the second subarea is training with multiple label as targets, and the third subarea is adversarial robustness. In the following, we explain the related work from all subareas.

### Normalization in DNNs

Normalization of the output is the most common use of internal manipulation in DNNs today. The most famous representative is the batch normalization(BN) (Ioffe and Szegedy 2015). This approach subtracts the mean and divides the output with the standard deviation, both are computed over several batches. In addition, the output is scaled and shifted by an offset. Those two values are also computed over several batches. Another type of output normalization is the group normalization GN (Wu and He 2018). In this approach, groups are formed to compute the mean and standard deviation, which are used to normalize the group. The advantage of GN in comparison to BN is that it does not require large batches. Other types of output normalization are instance normalization IN (Ulyanov, Vedaldi, and Lempitsky 2016; Huang and Belongie 2017) and layer normalization LN (Ba, Kiros, and Hinton 2016). LN uses the layers to compute the mean and the standard deviation, and IN uses only each instance individually. IN and LN are used in recurrent neural networks (RNN) (Schuster and Paliwal 1997)

or vision transformers (Dosovitskiy et al. 2020). The proposed tensor normalization belongs to this group, since we normalize the output of the rectifier linear units.

Another group of normalization modifies the weights of the model. As for the output normalization, there are several approaches in this domain. The first is the weight normalization (WN) (Salimans and Kingma 2016; Huang et al. 2017). In WN the weights of a network are multiplied by a constant and divided by the Euclidean distance of the weight vector of a neuron. WN is extended by weight standardization (WS) (Qiao et al. 2019). WS does not use a constant, but instead computes the mean and the standard deviation of the weights. The normalization is computed by subtracting the mean and dividing by the standard deviation. Another extension to WN is the weight centralization (WC) (Fuhl and Kasneci 2021) which computes a two dimensional mean matrix and subtracts it from the weight tensor. This improves the stability during training and improves the results of the final model. The normalization of the weights have the advantage, that they do not have to be applied after the training of the network.

The last group of normalization only affects the gradients of the models. The two most famous approaches are the usage of the first (Qian 1999) and second momentum (Kingma and Ba 2014). Those two approaches are standard in modern neural network training, since they stabilize the gradients with the updated momentum and lead to a faster training process. The main impact of the first momentum is that it prevents exploding gradients. For the second momentum, the main impact is a faster generalization. These moments are moving averages which are updated in each weight update step. Another approach from this domain is the gradient clipping (Pascanu, Mikolov, and Bengio 2012, 2013). In gradient clipping, random gradients are set to zero or modified by a small number. Other approaches map the gradients to subspaces like the Riemannian manifold (Gupta et al. 2018; Larsson et al. 2017; Cho and Lee 2017). The computed mapping is afterwards used to update the gradients. The last approach from the gradient normalization is the gradient centralization (GC) (Yong et al. 2020) which computes a mean over the current gradient tensor and subtracts it.

## Multi label image classification (MLIC)

In multi label image classification the task is to classify multiple labels correctly based on a given image. Since this is an old computer vision problem various approaches have been proposed here. The most common approach is ranking the labels based on the output distribution. This pairwise ranking loss was first used in (Usunier, Buffoni, and Gallinari 2009) and extended by weights to the weighted approximate ranking (WARP) (Gong et al. 2013; Weston, Bengio, and Usunier 2011). WARP was further extended by the multi label positive and unlabeled method (Kanehira and Harada 2016). This approach mainly focuses on the positive labels which have a high probability to be correct. This of course has the disadvantage that noisy labels have a high negative impact on the approach. To overcome this issue the top-k loss (Lapin, Hein, and Schiele 2015, 2016, 2017) was developed. For the top-k loss there are two representatives namely smooth top-k hinge loss and top-k softmax loss.

Another approach treats the multi label image classification problem as an object detection problem. The follow the two-step approach of the R-CNN object detection method (Girshick et al. 2014) which first detects possible good candidate areas and afterwards classifies them. The first approach in multi label image classification following this object detection approach is (Wei et al. 2015). A refinement of this approach is proposed in (Zhang et al. 2018; Nguyen et al. 2019) which uses an RNN on the candidate regions to predict label dependencies. The general disadvantage of the object detection based approach is the requirement of bounding box annotations. Similar to (Zhang et al. 2018; Nguyen et al. 2019) the authors in (Wang et al. 2016; Jin and Nakayama 2016) use a CNN for region proposal but instead of using only the candidate region, the authors use the entire output of the CNN in the RNN to model the label dependencies. Another approach which exploits semantic and spatial relations between labels only using image-level supervision is proposed in (Zhu et al. 2017). Another approach following the object detection problem concept uses a dual-stream neural network (Yu et al. 2019). The advantage is that the model can utilize local features and global image pairs. This approach was further extended by (Zhang et al. 2020) to also detect novel classes.

In the context of large scale image retrieval (Zhao et al. 2015; Lai et al. 2015) and dimensionality reduction (Mikalsen et al. 2019) the multi label classification problem also has an important share to the success. In (Zhao et al. 2015; Lai et al. 2015) deep neural networks are proposed to compute feature representations and compact hash codes. While these methods work effectively on multi class datasets like CIFAR 10 (Krizhevsky, Hinton et al. 2009) they are significantly outperformed on challenging multi-label datasets (Gordo et al. 2016). (Cevikalp, Elmas, and Ozkan 2016, 2018) proposed a hashing method which is robust to noisy labels and capable of handling the multi label problem. In (Kumar et al. 2019) a dimensionality reduction method was proposed which embeds the features and labels onto a low-dimensional space vector. (Mikalsen et al. 2019) proposed a semi-supervised dimension reduction method which can handle noisy labels and multi-labeled images.

## Adversarial Robustness

The most common defense strategies against adversarial attacks are adversarial training, defensive distillation and input gradient regularization. Adversarial training uses adversarial attacks during the training procedure or modify the loss function to compensate for input perturbations (Goodfellow, Shlens, and Szegedy 2014; Madry et al. 2017). The defensive distillation (Papernot et al. 2016) trains models on output probabilities and not on hard labels, as it is done in common multi class image classification.

Another strategy to train robust models is the use of ensembles of models (Strauss et al. 2017; Pang et al. 2019; He et al. 2017; Tramèr et al. 2017; Sen, Ravindran, and Raghunathan 2020). In (Strauss et al. 2017) for example, 10 models are trained and used in an ensemble. While those ensembles are very robust, they have a high compute and

memory consumption, which limits them to smaller models. To overcome the issue of high compute and memory consumption, the idea of ensembles of low-precision and quantized models has been proposed (Galloway, Taylor, and Moussa 2017). Those low-precision and quantized models alone have shown a higher adversarial robustness than their full-precision counterparts (Galloway, Taylor, and Moussa 2017; Panda, Chakraborty, and Roy 2019). The disadvantage of the low-precision and quantized models is the lower accuracy, which is increased by forming ensembles (Sen, Ravindran, and Raghunathan 2020). An alternative approach is presented in (Rakin et al. 2018), where stochastic quantization is used to compute low-precision models out of full-precision models with a higher accuracy and a high adversarial robustness.

## Method

In this paper, we present two optimizations for deep neural networks. One is the 2D tensor normalization and the other is the training of the full classification distribution and adaptation of the loss function. For this reason, we have divided the method part into two subgroups, in which both methods are described separately.

### Tensor Normalization

The idea behind the tensor normalization is to compensate the shifted value distribution after a rectifier linear unit. Since convolutions are computed locally, it is necessary that this normalization is computed for each $(x, y)$ coordinate separately. This results in a 2D matrix of mean values, which is subtracted from the tensor along the $z$ dimension.

$$TNMean_{x,y}(A) = \frac{\sum_{z=1}^{Z} A_{x,y,z}}{Z} \qquad (1)$$

Equation 1 describes the mean computation for the tensor normalization after the activation function. The tensor $A$ with the size $X, Y, Z$ is used online to compute the current 2D mean matrix $TNMean_{x,y}$ with the dimension $X, Y, 1$. Afterwards, this mean is subtracted from each $z$ position of the tensor and therefore, the entire tensor has a zero mean and a less skewed value distribution.

---

Algorithm 1: Algorithmic workflow of the tensor normalization in the forward pass. For the backward pass, the error values are simply passed backwards, since the subtraction equation in the derivative becomes 1.

---

**Data:** Activation tensor $A$
**Result:** Normalized activation tensor $A^*$
$M = TNMean(A)$
  **for** $i = 1; \ i < Z; \ i = i + 1$ **do**
  |  $A_i^* = A_i - M$
**end**

---

Algorithm 1 describes the computation of the tensor normalization in a neural network forward pass. As can be seen it is a simple online computation of the 2D mean matrix of the activation tensor and a subtraction along the depth of the tensor. For the backward pass the error values have just to be passed to the previous layer since the subtraction equation is one in the derivative. Due to this properties, it can be directly computed in the rectifier linear unit. This means it does not require any additional GPU memory.

*Our formal justification of "Why Tensor Normalization Improves Generalization of Neural Networks" is based on numerics and properties of large numbers. Mathematically, a neuron is a linear combination $P = D * W + B$ with $P$ =Output, $D$ =Input data, $W$ =Model weights, and $B$ =Bias term. If we now normalize our input data $A^* = (A - M)$ we get the formula $P = D^* * W + B$ with $M = Mean of D$. If we now simply define $B^* = B + M * W$, it follows that the normalization should have no effect on the neuron, since it can learn the same function even without the normalization. However, this changes when we consider the numerics and the computation of the derivatives in a neural network.*

*Suppose we have a one-dimensional input $D$ which is larger or equal than the normalized input $D^* = D - M$. The derivative for the weights is given by $\frac{\delta L}{\delta W} = \frac{\delta L}{\delta P} * \frac{\delta P}{\delta W} = (P - GT) * D$ with $L$ =Squared loss error function and $GT$ =Ground truth. As can be seen the data $D$ is included into the gradient computation of the weights which leads to larger steps in the error hyperplane. In addition, a large $D$ also results in smaller weights $W$ since $W = (P - B) * D^{-1}$. This means a large $D$ produces large gradient updates and searches for a smaller global optima $W$. With a smaller $D^* = D - M$ we look for a larger optima $W$ and use smaller gradient updates for this. In addition, the numerical stability of $W$ is higher since computers can only represent a certain accuracy for real numbers.*

*Proof that $|D^*| \leq |D|$: Since we apply the tensor normalization only after rectifier linear units $D \in \mathbb{R}_0^+$ and therefore $|D| \geq 0$, $|M| \geq 0$, and $|D^*| \geq 0$. Now we have to consider three cases $|D| = 0, |M| = 0$, $|D| > 0, |M| = 0$, and $|D| > 0, |M| > 0$. For the first case $|D| = 0, |M| = 0$, $|D^*|$ would also be zero and therefore $|D^*| \leq |D|$ holds. The second case $|D| > 0, |M| = 0$ leads to $D^* = D - M = D - 0 = D$ for which $|D^*| \leq |D|$ also holds. In the last case $|D| > 0, |M| > 0$ we can simply shift $M$ to the other side $D^* + M = D$ which shows that $|D^*| \leq |D|$ holds again.*

### Full Distribution Training

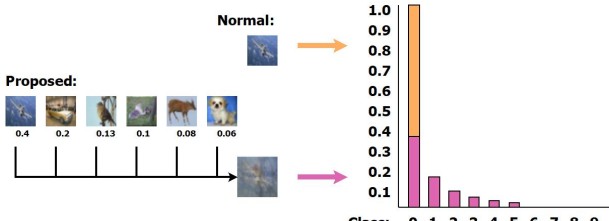

Figure 1: Exemplary illustration of the proposed full distribution training. In orange, the normal approach with one image to one class is shown. In pink, the combination of multiple images to one and the ground truth adaption is shown.

The idea behind the full distribution training is to not restrict the input to correspond to one class only. We combine multiple images using a weighting scheme and use this weighting as corresponding class labels. An example can be seen in Figure 1. For the computation of the weighting scheme, we use the harmonic series and select the amount of combined images randomly up to the amount of different classes. This makes it easier to reproduce our results and since the harmonic series is connected to the coupon collector's problem or picture collector's problem we thought it would be a superb fit. The purpose of the full distribution training is a cheap way to train robust models without any additional training time or specialized augmentation and maintaining the accuracy of the model.

$$F_i = \frac{\frac{1}{i}}{\sum_{j=1}^{max(C,RND)} F_j} \tag{2}$$

Equation 2 is the harmonic series ($\frac{1}{i}$) normalized by the sum ($\sum_{j=1}^{max(C,RND)} F_j$). We had to normalize the series because the harmonic series is divergent even though the harmonic series is a zero sequence. In Equation 2 $C$ represents the amount of classes of the used dataset and $RND$ a randomly chosen number.

$$D = \sum_{i=1}^{max(C,RND)} I_{j=RND} * F_i \mid C(j) \notin C(D) \tag{3}$$

With the factors from Equation 2 we can compute the new input images using Equation 3. Therefore, we multiply a randomly selected image $I_{j=RND}$ with the corresponding factor $F_i$ and combine all images by summing them up. However, there is a special restriction that only one example is allowed for each class ($C(j) \notin C(D)$). This means, that each class in $C(D)$ can only have one or no representative.

$$GT = \sum_{i=1}^{max(C,RND)} L_{j=RND} * F_i \mid C(j) \notin C(GT) \tag{4}$$

For the computation of the ground truth distribution $GT$ in Equation 4 we follow the same concept as for the images in Equation 3. We select the label $L_{j=RND}$ corresponding to the randomly selected image $I_{j=RND}$ and multiply it by the factor $F_i$. The combination is again done by summing all factorized labels together. As for the images, we allow only one example per class or none if the amount of combined images is less than the amount of classes.

The algorithmic description of the combination and weighting can be seen in Algorithm 2. In the first for loop we compute the factors, and in the second for loop we combine the images and the labels.

$$Softmax_i(P) = \frac{e^{P_i}}{\sum_{y=1}^{Y} e^{P_y}} \tag{5}$$

For the multi class classification, the softmax function has prevailed. The softmax function can be seen in Equation 5

**Algorithm 2:** The creation of a multi label example based on Equations 2, 3, and 4. In the first for loop the factors are computed and normalized. The second loop selects unique class examples and combines them based on the factors.

**Data:** Labels $L$, Images $I$, Classes $C$
**Result:** Ground Truth $GT$, Data $D$
$F = 0$;
$GT = 0$;
$D = 0$;
$Sum = 0$
$Amount = max(C, RND)$
**for** $i = 1$; $i < Amount$; $i = i + 1$ **do**
  $F_i = 1/i$
    $Sum = Sum + F_i$
**end**
$F = F/Sum$
  **for** $i = 1$; $i < Amount$; $i = i + 1$ **do**
  $j = RND(L) \mid C(j) \notin C(GT)$
    $GT = GT + L_j * F_i$
    $D = D + I_j * F_i$
**end**

and is used to compute an exponentially weighted distribution out of the predicted values. This distribution decouples the numeric values from the loss function so that only the relative value among the values is important, which stabilizes training and leads to a better generalization.

For the computation of the loss value and the back propagated error, Algorithm 3 is used in normal multi class classification. As can be seen in the first if statement, this is not sufficient for a multi label problem since we have multiple target values and those are not one ($P_S(y_i, b_i) = Scale * (P_S(y_i, b_i) - 1)$).

Therefore, we modified Algorithm 3 to Algorithm 4 which allows multiple labels with different values. This can be seen in the if condition ($GT(y_i, b_i) > \epsilon$) which handles all values greater $\epsilon$ and in the if branch ($P_S(y_i, b_i) = Scale * (P_S(y_i, b_i) - GT(y_i, b_i))$) which uses the ground truth value for gradient computation.

*Our formal justification that the full distribution training generates more robust networks: A common strategy to train more robust networks is the usage of Projected Gradient Descent (PGD), which for the sake of completeness is described in Section "Projected Gradient Descent (PGD)", during training. PGD computes the gradient of the current image and uses the sign of the gradient $sign(\delta f(x^t)$ to compute a new modified image $x^{t+1}$. This is done using an iterative scheme and an modification factor $\alpha$. The general equation for PGD is $x^{t+1} = x^t + \alpha * \delta f(x^t)$ whereby the $sign()$ function in Equation 6 is used to avoid that very small gradient values block the attack and feign robustness and also called $l_\infty$ norm. Our approach in contrast uses another image $I == x^0$ (or multiple images) from another class to modify the current image collection $D == x^{t+1}$. This means, that the gradient to shift one image into the direction of another class is gifted by the dataset itself through an image of another class. The modification equation for our approach*

Algorithm 3: The calculation of the softmax multi class log function, or also known as entropy loss. It first converts the predictions into a probability distribution using the softmax function. Afterwards, the desired class per batch gets the error based on its distance to 1 (if branch). All other values should be zero, which is why they receive their probability as error (else branch).

**Data:** Ground truth $GT$, predictions $P$, Batch size $B$
**Result:** Error $E$, Loss $L$
$P_S = Softmax(P)$;
  $Scale = \frac{1}{B}$;
  $L = 0$
  **for** $b_i = 1;\ b_i < B;\ b_i = b_i + 1$ **do**
  |  **for** $y_i = 1;\ y_i < Y;\ y_i = y_i + 1$ **do**
  |  |  **if** *if $y_i == GT(1, b_i)$* **then**
  |  |  |  $L = L + Scale * -log(P_S(y_i, b_i))$
  |  |  |  $P_S(y_i, b_i) = Scale * (P_S(y_i, b_i) - 1)$
  |  |  **else**
  |  |  |  $P_S(y_i, b_i) = Scale * (P_S(y_i, b_i))$
  |  |  **end**
  |  **end**
  **end**

---

*is $\sum_{i=1}^{max(C, RND)} I_{j=RND} * F_i \mid C(j) \notin C(D)$ based on Equation 3. If we set $max(C, RND) == 2$ we can remove the sum and get $I_{j1} * F_1 + I_{j2} * F_2 \mid C(j1) \neq C(j2)$. Now setting $F_1 == 1$ and $F_2 == \alpha$ we get $I_{j1} + \alpha * I_{j2} \mid C(j1) \neq C(j2)$. Since the class of $j1$ is different to the class of $j2$ we can interpret $I_{j2}$ as the gradient to another class and therefore write $I_{j2} = \delta f(I_{j1})$. With this gradient formulation we get $I_{j1} + \alpha * \delta f(I_{j1})$ which is the same as the PGD formulation. This means, that we can get our gradients to another class directly from the dataset and do not have to perform multiple iterations of forward and backward propagation to compute them. In addition, our approach can compute gradients into the direction of multiple classes.*

## Evaluation

In this section we show the numerical evaluation of the proposed approaches and describe the used datasets as well as the robust accuracy and PGD attack. For training and evaluation, we used multiple servers with multiple RTX2080ti or RTX3090 and cuda version 11.2. For the initialization of all networks, we use (He et al. 2015).

### Datasets

In this subsection all used datasets are described.

**CIFAR10** (Krizhevsky, Hinton et al. 2009) (C10) is a dataset consisting of 60,000 color images. Each image has a resolution of $32 \times 32$ and belongs to one of ten classes. For training, 50,000 images are provided and for training 10,000 images. Each class in the training set has 5,000 representatives and 1,000 in the validation set. Therefore, this dataset is balanced. *Data augmentation: Shifting by up to 4 pixels in each direction (padding with zeros) and horizontal flipping. Mean (Red=122, Green=117, Blue=104) subtraction as well as division by 256.*

Algorithm 4: The calculation of the softmax multi label log function, which we use for the full distribution training. It first converts the predictions into a probability distribution using the softmax function, as it is done in the softmax multi class log function. Afterwards, we use the ground truth distribution to select all classes in the current image ($GT(y_i, b_i) > \epsilon$) where $\epsilon$ is a small number greater zero. Based on the ground truth distribution value, we compute the error ($P_S(y_i, b_i) - GT(y_i, b_i)$). For all other values, we use the same procedure as in the softmax multi class log function (else branch).

**Data:** Ground truth $GT$, predictions $P$, Batch size $B$
**Result:** Error $E$, Loss $L$
$P_S = Softmax(P)$;
  $Scale = \frac{1}{B}$;
  $L = 0$
  **for** $b_i = 1;\ b_i < B;\ b_i = b_i + 1$ **do**
  |  **for** $y_i = 1;\ y_i < Y;\ y_i = y_i + 1$ **do**
  |  |  **if** *if $GT(y_i, b_i) > \epsilon$* **then**
  |  |  |  $L = L + Scale * -log(P_S(y_i, b_i))$
  |  |  |  $P_S(y_i, b_i) = Scale * (P_S(y_i, b_i) - GT(y_i, b_i))$
  |  |  **else**
  |  |  |  $P_S(y_i, b_i) = Scale * (P_S(y_i, b_i))$
  |  |  **end**
  |  **end**
  **end**

---

**CIFAR100** (Krizhevsky, Hinton et al. 2009) (C100) is a similar dataset in comparison to CIFAR10 but with the difference that it has one hundred classes. As in CIFAR10 each image has a resolution of $32 \times 32$ and three color channels. The amount of images in the training and validation set is identical to CIFAR10 which means that the training set has 50,000 images with 500 images per class. The training set has 10,000 images, with 100 images per class. Therefore, it is also a balanced dataset. *Data augmentation: Shifting by up to 4 pixels in each direction (padding with zeros) and horizontal flipping. Mean (Red=122, Green=117, Blue=104) subtraction as well as division by 256.*

**SVHN** (Netzer et al. 2011) consists of 630,420 images with a resolution of $32 \times 32$ and RGB colors. The dataset has 10 classes and is not balanced as the other datasets. The training set consists of 73,257 images, the validation set has 26,032 images, and there are also 531,131 images without label for unsupervised training. In our evaluation, we only used the training and validation set. *Data augmentation: Mean (Red=122, Green=117, Blue=104) subtraction as well as division by 256.*

**FashionMnist** (Xiao, Rasul, and Vollgraf 2017) (F-MNIST) is a dataset inspired by the famous MNIST (LeCun et al. 1998) dataset. It consists of 60,000 images with a resolution of $28 \times 28$ each. For training 50,000 images and for validation, 10,000 images are provided. Each image is provided as gray scale image, the dataset has 10 classes and is balanced as the original MNIST dataset. *Data augmentation: Shifting by up to 4 pixels in each direction (padding with zeros) and horizontal flipping. Mean*

Table 1: Comparison of the proposed approaches on multiple public datasets with the same preprocessing and learning parameters. OV represents the image manipulation of the full distribution training **without** the use of the adapted loss function (OV uses Algorithm 3). FDT is the full distribution training with the loss function from Algorithm 4. TN is the tensor normalization. Baseline is the accuracy without PGD, and $\epsilon$ represents the used clipping region for PGD. All results are the average over three runs, and $\pm$ indicates the standard deviation.
*Training parameters: Optimizer=SGD, Momentum=0.9, Weight Decay=0.0005, Learning rate=0.1, Batch size=100, Training time=150 epochs, Learning rate reduction after each 30 epochs by 0.1*
*Data augmentation: As statet in the dataset description section.*

| Dataset | Model | Baseline | $\epsilon = 10^{-1}$ | $\epsilon = 10^{-2}$ | $\epsilon = 10^{-3}$ | $\epsilon = 10^{-4}$ |
|---|---|---|---|---|---|---|
| C10 | ResNet-34 | $92.52 \pm 0.25$ | 6.28 | 54.90 | 91.93 | 92.51 |
| | ResNet-34 & OV | $92.13 \pm 0.37$ | 7.98 | 65.92 | 92.12 | 92.13 |
| | ResNet-34 & FDT | $93.13 \pm 0.19$ | 13.81 | 66.48 | 92.73 | 93.13 |
| | ResNet-34 & TN | $93.69 \pm 0.12$ | 5.85 | 54.75 | 91.72 | 93.69 |
| | ResNet-34 & TN & FDT | $\mathbf{93.77 \pm 0.20}$ | **14.75** | **68.53** | **93.01** | **93.76** |
| C100 | ResNet-34 | $73.16 \pm 0.61$ | 3.07 | 29.37 | 70.79 | 73.11 |
| | ResNet-34 & OV | $67.57 \pm 0.59$ | 3.89 | 36.17 | 66.39 | 67.57 |
| | ResNet-34 & FDT | $73.06 \pm 0.45$ | 6.06 | 42.69 | 72.12 | 73.06 |
| | ResNet-34 & TN | $\mathbf{74.80 \pm 0.22}$ | 3.90 | 33.64 | 70.81 | **74.72** |
| | ResNet-34 & TN & FDT | $74.37 \pm 0.27$ | **9.91** | **46.92** | **72.38** | 74.37 |
| F-MNIST | ResNet-34 | $96.1 \pm 0.23$ | 7.13 | 67.80 | 93.31 | 94.64 |
| | ResNet-34 & OV | $94.43 \pm 0.30$ | 34.16 | 87.87 | 93.82 | 94.43 |
| | ResNet-34 & FDT | $96.01 \pm 0.26$ | 36.48 | **88.51** | 94.50 | 95.92 |
| | ResNet-34 & TN | $\mathbf{96.46 \pm 0.14}$ | 9.50 | 74.90 | 93.76 | 94.70 |
| | ResNet-34 & TN & FDT | $96.13 \pm 0.22$ | **39.03** | 86.54 | **94.93** | **95.94** |
| SVHN | ResNet-34 | $94.83 \pm 0.22$ | **18.64** | 82.77 | 91.01 | 94.79 |
| | ResNet-34 & OV | $94.13 \pm 0.35$ | 5.82 | 50.23 | 93.14 | 94.13 |
| | ResNet-34 & FDT | $95.01 \pm 0.21$ | 12.87 | 77.62 | 92.09 | 95.01 |
| | ResNet-34 & TN | $\mathbf{95.21 \pm 0.18}$ | 17.02 | **83.73** | **95.21** | **95.21** |
| | ResNet-34 & TN & FDT | $95.16 \pm 0.16$ | 18.05 | 82.04 | 94.73 | 95.16 |

(Red=122, Green=117, Blue=104) subtraction as well as division by 256.

## Projected Gradient Descent (PGD)

To evaluate the robustness of the models, we use the widely used PGD (Madry et al. 2017) method. Here, the gradient is calculated for the current image and iteratively applied to the image to manipulate it and cause misclassification.

$$x^{t+1} = Clip_{-\epsilon,\epsilon}(x^t + \alpha * sign(\delta f(x^t))) \qquad (6)$$

Equation 6 shows the general equation of PGD and $x^0$ is the original input image. $x^{t+1}$ is the computed input image for this iteration, $Clip_{-\epsilon,\epsilon}$ is a function to keep the image manipulation per pixel in the range $-\epsilon$ to $\epsilon$, $x^t$ is the image from the last iteration, $\alpha$ is the factor which controls the strength of the applied gradient, and $sign(\delta f(x^t))$ is the gradient sign per pixel of the current input image $x^t$. The $sign()$ function corresponds to the $l_\infty$ norm and is the strongest PGD based attack since the value of the gradient has no influence to the perturbation but only the sign.

In our evaluation we set the maximum amount of iterations $T = 40$, $\alpha$ was initialized with $\alpha = \epsilon * \frac{0,01}{0,3}$ as it is done in Foolbox (Rauber, Brendel, and Bethge 2017) and evaluated the $\epsilon$ in the range of 0.1 to 0.0001.

$$Accuracy = \frac{\sum_{x_i^0}^{X^0} \sum_{t=1}^{T} C(f(x_i^t)) == C(x_i^0)}{|X^0| * T} \qquad (7)$$

Equation 7 shows the computation of the robust accuracy in this paper with the dataset $X^0$, the single images $x_i^0$, the amount of iterations $T$, the model $f()$, and the ground truth class $C()$. This is the same computation as it is done for the normal image classification task, but with the difference that each perturbation of the input image is counted separately.

## Evaluation of the Tensor Normalization (TN) and Full Distribution Training (FDT)

All results with a ResNet-34 on the CIFAR 10, CIFAR 100, Fashion Mnist, and SVHN datasets can be seen in Table 1. Comparing the baseline results, it is evident that tensor normalization (TN) outperforms all other combinations. However, the full distribution training (FDT) also improves the results, which is mainly due to the multi label variant of the loss function and the reformulation to a multi label problem (Uses Algorithm 4). This is especially obvious by the comparison of FDT to OV (Uses Algorithm 3). If OV is considered, it can be seen that the superposition of multiple images improves the robustness, but also has a negative impact on the accuracy of the model. Comparing the robustness of the models for $\epsilon = 10^{-1}$, one can clearly see that FDT increases

Table 2: Evaluation of the proposed methods on larger DNN model in comparison to the vanilla version. Baseline is the accuracy without PGD and $\epsilon$ represents the used clipping region for PGD.
*Training parameters: Optimizer=SGD, Momentum=0.9, Weight Decay=0.0005, Learning rate=0.1, Batch size=100, Training time=150 epochs, Learning rate reduction after each 30 epochs by 0.1*
*Data augmentation: As statet in the dataset description section.*

| Dataset | Model | Baseline | $\epsilon = 10^{-1}$ | $\epsilon = 10^{-2}$ | $\epsilon = 10^{-3}$ | $\epsilon = 10^{-4}$ |
|---------|-------|----------|----------|----------|----------|----------|
| C100 | ResNet-152 | 76.09 | 3.13 | 28.97 | 71.05 | 75.96 |
| | ResNet-152 & FDT & TN | **77.11** | **10.28** | **50.09** | **74.12** | **77.01** |
| | WideResNet-28-10 | 78.23 | 4.57 | 32.50 | 73.58 | 77.91 |
| | WideResNet-28-10 & FDT & TN | **79.06** | **13.59** | **54.34** | **75.68** | **78.98** |

the robustness significantly as well as the combination of TN and FDT brings a further improvement. **What is also notable are the results for SVHN, here FDT does not seem to have a positive impact on the robustness of the models. This is due to the fact that the images in SVHN already contain several classes (house numbers) and only the middle one is searched. Therefore, the multi label reformulation is not entirely valid since gradients from multiple classes are already present, which can be seen in the results of the robust accuracy. Looking at the result for $\epsilon = 10^{-1}$ of the vanilla ResNet-34 for the dataset SVHN, one sees directly that this is already very robust. Since there are multiple house numbers in each image, this follows the approach of OV. Since this is only true for the SVHN dataset and all other datasets become significantly more robust using FDT, this confirms the basic idea of our approach of using single images from different classes to generate gradients pointing to other classes.** The fact that OV does not become more robust for SVHN can be explained by the fact that it represents an exaggerated data augmentation, which can be seen in the worst overall accuracy as well as the susceptibility to PGD.

For all models, we used the same parameters as well as the same number of epochs for training. It is interesting to note here that FDT and TN can thus be used in the same time and with the same number of learnable parameters. For TN, however, it is important to note that this operation represents an additional computational cost, whereas the calculation of the 2D mean matrix and the subtraction do not represent a significant difference in execution time, nor an increase in the complexity of the model.

Table 2 shows the results of full distribution training and tensor normalization on CIFAR 100 with large models compared to the vanilla version. As can be seen, both approaches improve the accuracy of the model and the robust accuracy by more than twice of the vanilla version for $epsilon = 10^{-1}$. Considering that no further parameters and no further training time are needed, this is a significant improvement, as seen by the authors.

## Conclusion

In this paper, we have presented a novel approach to train deep neural networks that converts the multi-class problem into a multi-label problem and thereby generates more robust models. We name this approach full distribution training and used the harmonic series for the generation of the labels as well as for the image combination. This series can be replaced by any other series or just by random factor selection but would require an immense amount of evaluations which is out of the scope of this paper as well as incredibly harmful to nature since GPUs require a large amount of energy. Additionally, we have algorithmically presented the reformulation of the multi class loss function into a multi label loss function and formally justified the functionality of this reformulation. In addition to the reformulation, we introduced and formally described tensor normalization and formally showed that it will improve the results. All theoretical conjectures were confirmed by evaluations on multiple publicly available datasets for small ResNet-34 as well as two large DNNs (WideResNet-28-10 and ResNet-152).

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
