# OpenReview forum: "Tensor Normalization and Full Distribution Training"
_AAAI.org/2022/Workshop/AdvML — AAAI-22 AdvML Workshop LongPaper_

### Official Review · Reviewer_b7Y4 · 2021-11-27
**Review of paper "Tensor Normalization and Full Distribution Training"**

**Rating:** 6
**Confidence:** 4

**Review:**

This paper proposes two techniques including tensor normalization and full distribution training to improve model robustness. These two techniques are easy to understand, and bring improved robustness compared with the baseline. Here are some suggestions for the authors.

- Can the proposed methods be integrated with adversarial training? And how about the results?
- Do the proposed methods have other advantages beyond adversarial robustness, such as natural robustness evaluated by CIFAR-C.

Hope the authors can further improve this paper.

---

### Official Review · Reviewer_KVLA · 2021-11-30
**A new normalization method.**

**Rating:** 5
**Confidence:** 3

**Review:**

The paper introduces a new normalization method named pixel-wise tensor normalization which improves both accuracy and robustness of the model. However, the results shows somewhat improvement, but not significant. Also, I think the paper is not providing enough theoretical backups for the claimed algorithm, and it prevents me from being completely convinced. Also, the paper does not seem to be a complete draft - there are many points that seem to be incomplete. The paper still needs further polishment and is not ready for publication at the moment.

---

### Decision · Program_Chairs · 2021-12-01

**Decision:**

Accept (Long Paper)

**Comment:**

Reviewer KVLA raises some concerns on the completeness of this paper, suggesting that this paper needs further improvements for publication. Based on the comments of both reviewers, this paper is accepted as a long paper. Please address the reviewers' comments in the final version.